# SH-29 and SK-119 Attenuates Air-Pollution Induced Damage by Activating Nrf2 in HaCaT Cells

**DOI:** 10.3390/ijerph182312371

**Published:** 2021-11-24

**Authors:** Shirin Kahremany, Lukas Hofmann, Noy Eretz-Kdosha, Eldad Silberstein, Arie Gruzman, Guy Cohen

**Affiliations:** 1Department of Chemistry, Faculty of Exact Sciences, Bar-Ilan University, Ramat-Gan 5290002, Israel; shirin.kahremany@live.biu.ac.il (S.K.); lukas.hofmman@biu.ac.il (L.H.); 2The Dead Sea and Arava Science Center, The Skin Research Institute, Masada 8691000, Israel; noy@adssc.org; 3Department of Plastic Surgery, Soroka University Medical Center, Ben-Gurion University of the Negev, Beer-Sheva 8410100, Israel; eldads@bgu.ac.il; 4Eilat Campus, Ben Gurion University of the Negev, Eilat 8855630, Israel

**Keywords:** Nrf2, diesel particulate matter (DPM) (E)-5-oxo-1-(4-((2,4,6-trihydroxybenzylidene)amino)phenyl)pyrrolidine-3-carboxylic acid (SK-119), 2,2′-((1E,1′E)-(1,4-phenylenebis(azaneylylidene))bis(methaneylylidene))bis(benzene-1,3,5-triol (SH-29)), keratinocytes, pharmacological activators, ROS

## Abstract

Air pollution has been repeatedly linked to numerous health-related disorders, including skin sensitization, oxidative imbalance, premature extrinsic aging, skin inflammation, and increased cancer prevalence. Nrf2 is a key player in the endogenous protective mechanism of the skin. We hypothesized that pharmacological activation of Nrf2 might reduce the deleterious action of diesel particulate matter (DPM), evaluated in HaCaT cells. SK-119, a recently synthesized pharmacological agent as well as 2,2′-((1E,1′E)-(1,4-phenylenebis(azaneylylidene))bis(methaneylylidene))bis(benzene-1,3,5-triol) (SH-29) were first evaluated in silico, suggesting a potent Nrf2 activation capacity that was validated in vitro. In addition, both compounds were able to attenuate key pathways underlying DPM damage, including cytosolic and mitochondrial reactive oxygen species (ROS) generation, tested by DC-FDA and MitoSOX fluorescent dye, respectively. This effect was independent of the low direct scavenging ability of the compounds. In addition, both SK-119 and SH-29 were able to reduce DPM-induced IL-8 hypersecretion in pharmacologically relevant concentrations. Lastly, the safety of both compounds was evaluated and demonstrated in the ex vivo human skin organ culture model. Collectively, these results suggest that Nrf2 activation by SK-119 and SH-29 can revert the deleterious action of air pollution.

## 1. Introduction

Despite significantly improved air quality in high-income countries due to agreements such as the Clean Air Act, air pollution is still one of the major environmental health risks [1]. Chronic or repeated exposure to pollutants has been repeatedly associated with premature deaths and the loss of millions of disability-adjusted life-years around the world. Indeed, the European health agency still considers air pollution as the largest environmental health risk, ultimately causing cardiovascular and respiratory diseases that might lead to premature deaths [2,3,4]. Environmental pollutants such as polycyclic aromatic hydrocarbons, volatile organic compounds, heavy metals, ozone (O_3_), cigarette smoke, and particulate matter (PM) were repeatedly linked to respiratory and systemic disorders as well as an increase in the prevalence of cutaneous inflammatory diseases [5,6,7]. For instance, PM in ambient air is strongly related to the progression of atopic dermatitis in children. In addition, high levels of PM intensify their symptoms, such as itching, exacerbating of pre-existing disease (reviewed in [8,9]). Other inflammatory diseases such as psoriasis, contact dermatitis, and allergic manifestations were also linked to air pollution [10,11]. Skin sensitization, premature skin aging, and even cancer prevalence have been suggested to be intensified by pollutants [12].

Although the skin act as a physical, chemical, and biological barrier, several studies showed that environmental pollutants concentrating at its surface might affect the skin. In addition, airborne uptake was demonstrated [13]. However, the detailed mechanism of how air pollution leads to inflammatory skin disorders is not fully understood. It is believed that air pollution triggers a cascade of oxidative stress by accumulating on the skin surface leading to absorption through hair follicles or inhalation and diffusion into deeper dermal layers [13]. Once these pollutants enter the skin, they cause local inflammation, which increases the amount of ROS. In the first step, the production of ROS will induce activation of the antioxidant defense system. However, at a higher concentration of chronic exposure, the increased stress will induce oxidative stress and inflammatory response and may lead to the collapse of the defense system and the development of tissue damage.

The nuclear factor erythroid 2-related factor 2 (Nrf2) signaling pathway is crucial for the endogenous defense system of the skin and enables the tissue to cope with the excess of concomitant damage and ROS [14]. Moreover, the Nrf2 signaling pathway has protective effects against air pollutants [15]. Nrf2, ARE, and Kelch-like ECH-associated protein 1 (Keap1) are the three pillars of this pathway and facilitate the proper functioning of this critical antioxidant defense mechanism. Normally, two Keap1 molecules and one Nrf2 molecule form a heterotrimer that continuously degrades via proteasomes [16,17,18]. Under oxidative stress, Nrf2 is released from Keap-1, leading to activation of the Nrf2 pathway and subsequently expression of ARE-dependent genes, which ultimately balance oxidative mediators.

An increasingly prevalent strategy is upregulating Nrf2 to competitively disrupt the protein–protein interaction (PPI) between the Keap1 Kelch domain and the Nrf2 Neh2 peptide motifs [19]. We have recently applied an in silico approach in order to design and synthesize a novel Nrf2 enhancer, (E)-5-oxo-1-(4-((2,4,6-trihydroxybenzylidene)amino)phenyl)pyrrolidine-3-carboxylic acid (SK-119). Our data demonstrated that SK-119-treated cells and tissues displayed a reduction in cytokine secretion induced by lipopolysaccharides comparable to the effects of dexamethasone. In addition, topical application of SK-119 was able to block UVB-induced oxidative stress and attenuated caspase-mediated apoptosis, DNA adduct formation, and concomitant cellular damage.

The current research was aimed to attenuate air-pollution-induced damage by increasing the endogenous protective mechanism of the skin by pharmacological activation of Nrf2. SK-119 and the novel molecule SH-29 designed in silico were evaluated in diesel particulate matter (DPM)-induced damage in vitro.

## 2. Materials and Methods

All chemical reagents, solvents, and acids were purchased from Acros Organic (Yehud, Israel), Alfa Aesar (Ward Hill, MA, USA), Bio-Lab Ltd. (Jerusalem, Israel), Merck (Rehovot, Israel), or IUCHEM Ltd. (Shanghai, China), and all were used as received. Analytical and preparative HPLCs (Young Lin Instruments, Anyang, Korea) were performed on LUNA C18 preparative (10 µm, 100 mm × 30 mm) or analytical (5 µm, 250 mm × 4.6 mm) columns, both from Phenomenex Inc. (Torrance, CA, USA). HPLC purification was carried out with an increasing linear gradient of CH_3_CN in H_2_O. The purity of the synthesized compounds was confirmed by HPLC analysis. Analytical TLC was carried out on precoated silica gel 60 F_254_ (Merck) sheets using UV absorption and iodine physical adsorption for visualization. Mass spectra were recorded on a Finnigan model 400 instrument using a QToF microspectrometer (Micromass, Milford, MA, USA) by electrospray ionization in the positive/negative ion modes. The data were processed using mass LynX ver. 4.1 calculations and deconvolution software (Waters Corp., Milford, MA, USA). High-resolution mass spectra were obtained using an LTQ Orbitrap XL (Thermo Scientific, Waltham, MA, USA) mass spectrometer. The melting point of SK-119 was measured with a Fisher-Johns melting point apparatus (Waltham, MA, USA). All cell culture media and reagents were purchased from Biological Industries Ltd. (Beit-HaEmek., Israel). Diesel particulate matter (DPM) was purchased from Sigma-Aldrich Israel (by the National Institute of Standards and Technology (NIST), MD, USA). DPM stock solution (50 mg/mL in ethanol) was freshly prepared on the day of each experiment and mixed vigorously.

### 2.1. Computational Studies

#### 2.1.1. Ligand Preparation

The structures were prepared using the LigPrep module in the Schrodinger suite (Maestro12.7). Using an OPLS_2005 force field, the LigPrep produces energy minimized 3D structures. For each structure, the tautomer, correct Lewis structure, and ionization states (pH 7.0 ± 2.0) were generated, optimized, and energy minimized under default settings.

#### 2.1.2. Conformational Search

Conformers of each molecule were generated in MacroModel using the OPLS_2005 force field, GB/SA water, and no cutoff for nonbonded interactions. Molecular energy minimizations were performed using the PRCG method with 5000 maximum iterations and 0.05 gradient convergence threshold. The conformational searches were carried out by application of the Mixed torsional/Low-mode sampling method, performing automatic setup with 21 kJ/mol in the energy window for saving structure and a 0.5 Å cutoff distance for redundant conformers.

#### 2.1.3. Protein Preparations and Receptor Grid Generation

The crystal structure of Keap1 (PDB code: 2FLU) was imported into the maestro workspace, and the multistep Protein Preparation Wizard was used to correct the protein structure, which includes the addition of H-atoms, bond order correction, and H-bond network optimization, followed by energy minimization using OPLS_2005 force field. After preparation, the receptor grid was generated with Glide by specifying the binding site with a 3D cubic box. A box of 20 Å side lengths was placed for the enclosure. Potential energies of receptor atoms confined in this box were calculated without scaling their Van der Waals radius.

#### 2.1.4. Molecular Docking

##### Glide

The molecular docking was performed with the Glide XP (Xtra Precision) protocol of the Schrodinger Suite with default settings [20]. Default settings were used. Docking boxes were centered on the identified site, and Glidescore was used to evaluate the resulting poses.

##### Induced Fit Docking

Induced fit docking (IFD), which allows both receptor side chain and ligand flexibility, were carried out. IFD was performed to determine the interaction potential of SK-119 and SH-29 to Keap1 when receptor-side chain flexibility is induced and no constraints are given. The receptor grid parameters were kept the same as Glide docking. Other internal settings and parameters in induced fit workflow were kept at their default states and values.

#### 2.1.5. Binding Free Energy Analysis for Drug-Target Binding Energy Estimation

The interaction energies between the target protein and the selected top poses were computed using the MM-GBSA (generalized-born/surface area) method implemented in Schrodinger [21]. This binding energy estimates the stability of the protein complex. Prime MM-GBSA is a physics-based method that calculates the force field energies of the bound and unbound states of the protein–ligand complex [22]. The Van der Waals surface-based surface generalized born implicit solvation model was used with the OPLS_2005 force field, and residue flexibility was defined throughout the structure. The MM-GBSA binding free energy is defined as follows: [23]
(1)$$\mathrm{MM}-\mathrm{GBSA}\;\Delta G=\Delta E_{mm}+\Delta G_{sol}+\Delta G_{sa}$$

Δ*E**_mm_* is the difference in energy between the complex structure and the sum of the energies of the unbound ligand and protein. Δ*G**_sol_* is the difference in the solvation energy of the complex and the sum of the solvation energies for the unbound ligand and protein. Δ*G**_sa_* is the difference in the surface area energy for the complex and the sum of the surface area energies for the unbound ligand and protein [24]. Entropy terms related to the ligand and protein are not incorporated due to the expensive computational demand and no apparent improvement in many cases [25].

#### 2.1.6. HOMO LUMO Stability Analysis of the Screened Compounds

Density functional theory (DFT), a quantum chemistry method, was applied to investigate the detailed aspects in terms of structure, electronics, and energy states of every atom of ligand. For this study, HOMO, LUMO, and MESP analysis were performed on both SK-119 and SH-29 using B3LYP (Lee–Yang–Parr correlation functional theory) incorporation of basis set 6-31G** level and hybrid DFT with Becke’s 3-parameter exchange potential [26]. The gap energy is defined as the difference between HOMO and LUMO molecular orbital energy; energy indicates the excitation energy as well as showing the stability and reactivity of the compounds. The outcome of all optimized structures was examined in the form of molecular frontier orbital’s MESP, HOMO, and LUMO using Maestro panel, Schrodinger. The MESP *V(r)* at point r, due to a molecular system with nuclear charges [26], located at [23] and the electron density (*r*), was derived using the following equation:(2)$$v\left(r\right)=\;\overset{N}{\underset{A=1}{\sum }}\frac{Z_{A}}{\left|r-R_{A}\right|}-\int \frac{\rho \left(r^{\prime }\right)d^{3}r^{\prime }}{|r-r^{\prime }|\;}\;$$
where *N* indicates the total number of nuclei in the molecule, and the two terms refer to the bare nuclear potential and the electronic contributions, respectively. Molecular frontier orbitals HOMO and LUMO, as well as MESP of all optimized structures, were visualized with Maestro 12.7. The color-coded isosurface of MESP maps affords the overall size of the molecule and the locations of negative and positive electrostatic potentials.

### 2.2. Synthetic Chemistry

2,2′-((1E,1′E)-(1,4-phenylenebis(azaneylylidene))bis(methaneylylidene))bis(benzene-1,3,5-triol) (SH-29). Compound **1** and 2,4,6-trihydroxybenzaldehyde (both 2 mmol) were dissolved in ethanol (10 mL) and left overnight at room temperature. Ethanol was evaporated and after recrystallization from acetonitrile, and SH-29 was obtained (Brown-red powder, yield 86%, melting point >300 °C) ^1^H NMR (400 MHz, DMSO-d6): δ 10.27 (br s, 1H), 8.94 (s, 2H), 7.38 (s, 4H), 5.89 (s, 4H) ppm. ^13^C NMR (100 MHz, DMSO-d6) δ 167.3, 164.5, 162.6, 156.6, 120.9, 119.6, 94.2 ppm. MS found (M − H)^−^ (*m*/*z*) 379.093; calculated for C_20_H_16_ N_2_O_6_: *m*/*z*, 380.

5-oxo-1-(4-((2,4,6-trihydroxybenzylidene)amino)phenyl)pyrrolidine-3-carboxylic acid (SK-119). The compound was prepared according to the procedure described in our previous work [27]. Briefly, itaconic acid and phenyldiamine were mixed in water in a ratio of 1:0.9 for 1 h under reflux. Next, the compound was purified using several extractions and purification steps. First, using water at basic pH, unreacted itaconic acid and the desired compound **3** were separated from the leftovers of the phenyldiamine. In the second step, by adding water at acidic pH, the itaconic acid and compound **3** were separated. Finally, using preparative HPLC, compound **3** was purified. Next, compounds **3** and 2,4,6-trihydroxybenzaldehyde (both 2 mmol) were dissolved in ethanol (10 mL) and left overnight at room temperature. Ethanol was evaporated and after recrystallization from the same solvent, SK-119 was obtained (red powder, yield 39%, melting point >300 °C) ^1^H NMR (400 MHz, DMSO-d6): δ 10.16 (br s, 1H), 8.92 (s, 1H), 7.70 (d, J = 8.80 Hz, 2H), 7.34 (d, J = 8.80 Hz, 2H), 5.84 (s, 2H), 4.03 (m, 2H), 3.38 (m, 1H), 2.76 (m, 2H) ppm. ^13^C NMR (100 MHz, DMSO-d6) δ 174.2, 171.7, 164.1, 164.5, 156.7, 143.5, 137.2, 121.0, 120.3, 101.4, 94.2, 50.0, 35.2, 35.1 ppm. MS found (M + H)^+^ (*m*/*z*) 357.1; calculated for C_18_H_16_N_2_O_6_ *m*/*z*, 356.10.

### 2.3. Experimental Biology

#### 2.3.1. Cell Culture and Ex Vivo Skin Cultures

The human keratinocyte cell line (HaCaT) was purchased from CLS Cell Lines Service GmbH. The cells were grown in DMEM 4.5 g/L glucose supplemented with 10% fetal bovine serum and 1% (*v*/*v*) penicillin/streptomycin and maintained at 37 °C in a humidified 5% CO_2_ incubator. Unless specified, the cells were always seeded at 150,000 cells/mL and used the following day. Skins were obtained from 40 to 60-year-old healthy women undergoing aesthetic abdomen surgery after signing an informed consent form. The experiments were conducted with the approval of the IRB (Helsinki Committee) of Soroka Medical Center, Beer Sheva, Israel (#0258-19-SOR, approval protocol scrc20016; 29 June 2020). The study was initiated on the day of surgery. Skin culturing and treatment was performed under sterile conditions. A mechanical press was used to section the skin into 0.8 × 0.8 cm^2^ pieces, as previously described [16]. The skin explants were maintained in an air/liquid interface with the dermal side submerged in the medium, as described [28]. All compounds were applied topically on the epidermal layer.

#### 2.3.2. Cell and Tissue Viability Assay

In order to verify that the treatments were non-toxic, the MTT assay was employed. Following treatments, the cell culture media was aspirated, and 0.5 mg/mL MTT (3-(4,5- dimethylthiazol-2-yl)-2,5-diphenyltetrazolium bromide dissolved in PBS) were added. After 1 h incubation at 37 °C, the solution was aspirated, and isopropanol was added to the cells in order to solubilize the colored crystals. The absorbance at 570 nm was measured in an ELISA reader. Viability in the ex vivo skin was performed as described before [29]. Briefly, in order to detach the epidermis, the skin tissues were placed in PBS (56 °C) for 1 min. Then, the epidermal layers were easily removed by gentle application of a scalpel. The epidermis sheets were then placed in a 96-well plate containing the MTT solution. Following 1 h incubation at 37 °C, the epidermal sheets were transferred to a new well containing isopropanol, and the eluted dye absorbance was measured by plate reader.

#### 2.3.3. Nrf2-Activation Assay

After 2 h incubation without or with the compounds, the nuclear fraction was isolated using a nuclear extract kit (Active Motif, Carlsbad, CA, USA). Following protein defemination, 30 μg of each sample was taken incubated for 1 h in a 96-well plate coated with ARE sequence oligonucleotide. The bound Nrf2 was captured by anti-Nrf2 antibody and visualized by colorimetric reaction using secondary antibodies (TransAM Nrf2 DNA; Active Motif). The resultant color was measured spectrophotometrically at 450/655 nm.

#### 2.3.4. In Vitro Antioxidant Activity

The direct scavenging activity of the compounds was determined by the DPPH (2,2-Diphenyl-1-picrylhydrazyl) method. DPPH reagent was dissolved in ethanol to yield a 290 µM working solution that was freshly prepared on the day of the experiment. The compounds (final volume of 20 µL) were added to 380 µL of DPPH working solution. Concomitantly, Trolox was added similarly to the DPPH working solution in order to generate the calibration curve. After 10 min incubation period at RT in reduced light condition, the absorbance of 100 µL aliquots was measured by spectrophotometer (517 nm). The scavenging efficacy is determined in the linear range of the curve, as written below:(3)$$\mathrm{Scavenging}\;\mathrm{efficacy}\;=\frac{\left(O.D.\;\mathrm{vehicle}\;-\;O.D.\;\mathrm{treatment}\right)\;*\;100}{O.D.\;\mathrm{vehicle}}$$

#### 2.3.5. Intracellular and Mitochondrial Reactive Oxygen Species (ROS) Determination

The cells were seeded in 96 well plates at 150,000 cells/mL in a final volume of 170 µL of complete growth media and incubated for 24 h. Then, the medium was aspirated, and the cells were loaded with 50 µM dye 5, (and 6)-carboxy-2′ 7′-dichlorodihydrofluoresceine diacetate (DC-FDA) cytosolic ROS indicator or MitoSOX for monitoring mitochondrial ROS formation. Then, the cells were incubated at 37 °C with DPM (in PBS) at a final concentration of 100 µg/mL in the absence or presence of the compounds (tested at 1–100 µg/mL). Fluorescence was determined (ex. 485 nm, em. 538 nm) after 45 min of incubation.

#### 2.3.6. IL-8 and IL-1α Quantification

Following treatment, the spent media of the cell culture or skin organ culture was collected, cleared by centrifugation (1500 rpm, 5 min.), aliquoted, and stored at −80 °C until used. Human IL-8 or IL-1α quantification was performed by ELISA kits (sensitivity of 8 pg/mL), according to the manufacturer’s instructions and using standard curves. Absorbance was measured at the wavelength 540 nm using a microplate reader (Infinite f200, TECAN).

#### 2.3.7. Histology

The ex vivo skin tissues were fixed with 4% formaldehyde for 24 h at 4 °C, washed with PBS, and kept at 70% ethanol until use. Following dehydration in gradual increasing concentrations of ethanol and embedment, 8 µm paraffin sections were prepared, pasted and the slides were stained with hematoxylin eosin (H&E). Pictures were captured by using the Zeiss Inverted microscope Axio Observer 7 and the Moticam 5 + Camera.

### 2.4. Statistical Analysis

Values are presented as the average of three experiments, and standard errors of the mean (SEM) are provided. Significant differences between values were analyzed using the unpaired t-test, while significant results are for *p* < 0.05.

## 3. Results

The synthesized compound SH-29 was designed based on the structure of SK-119 [27]. These two compounds differ in polyphenolic rings; SK-119 has one triphenolate ring, whereas compound SH-29 has a double triphenolate ring. Compound SH-29 was obtained by conjugation between p-phenylenediamine and 2,4,6-trihydroxybenzaldehyde (Appendix A). In our previous in silico studies, SK-119 was shown to interact with the arginine-rich area in the Kelch domain of Keap1 using the same mode as the Nrf2-ETGT motif through hydrophobic and electrostatic interactions: two π–π stacking interactions between the compound and protein side chains (TYR 334 and TYR 572) and several H-bonds with the side chains (ARG 415, ARG 483, SER 508, SER 555, and TYR 334). It was also shown that SK-119 was able to attenuate skin inflammatory and UVB-induced damage by Nrf2 activation. Here we performed an extensive in silico calculation with our novel compound SH-29 and compared its properties with SK-119. Figure 1 describes the computational workflow.

### 3.1. Molecular Docking

Compound SH-29 and SK-119 were subjected to a conformational search procedure prior to docking. The conformations were subsequently subjected to XP docking using Glide, the dock poses were ranked according to the docking score, and the top 20 poses were selected as a starting point for Induced fit docking (IFD). IFD was carried out to study their binding poses, interact with the receptor, and predict binding affinity in terms of docking score, while both receptor-side chain and ligand atoms were flexible. The docked conformation with the lowest docking score was selected to analyze the binding pattern. The results show that both compounds had no interactions with any of the key amino acids mentioned above. However, they bind in a different binding mode to the binding site of Keap1, as shown in Figure 2. Compound SH-29 was found to have a higher docking score and forms H-bond interaction with residues SER 602, ASN 414, ARG 415, GLN 530, TYR 525, and π–π stacking with PHE 572, TYR 525 and hydrophobic interaction with TYR 334, PHE 577, ALA 556, TYR 572, TYR 525, GLY 364, GLY 603, GLY 574 with docking score −6.571. However, compound SK-119 forms H-bond interactions with residues ARG 483, ARG 415, and π–π stacking with PHE 572 and hydrophobic interactions with TYR 334, PHE 577, TYR 572, TYR 525, GLY 509 with docking score −6.382. (Figure 2).

### 3.2. Calculation for Prime Molecular Mechanics/Generalized Born Surface Area (MM-GBSA)

Post-scoring approaches of docking, namely MM/GBSA (binding free energy), were calculated for SH-29 and SK-119 to verify the precision of the docking protocol. The total binding free energy was contributed from the Coulomb energy (Coulomb), the covalent binding (Covalent), the hydrogen bonding (H-bond), the lipophillic binding (Lipo), the π–π packing interaction (Packing), the solvent generalized binding, and the binding from the Van der Waals energy (VDW). The contribution of each item is indicated in Figure 3. The binding free energy value was −54.58 kcal/mol for compound SH-29 and −49.48 kcal/mol for compound SK-119. Compound SH-29 showed lower total binding free energy. Among all the interactions, the contribution of the lipophilic binding and the Van der Waals energy was more than other items. All energies are represented in Table 1 and Figure 3.

### 3.3. Molecular Electrostatic Potential (MESP), and HOMO LUMO Stability Analysis of SK-119 and SH-29

The electronic and energetic states of both compound SH-29 and compound SK-119 were examined and represented in the form of MESP, HOMO, and LUMO in Figure 4. For both compounds, SH-29 and SK-119, the HOMO’s outcome ranges from −0.127 to −0.222 and LUMO −0.014 to −0.113. The orbital energy and the difference between HOMO and LUMO energy (gap energy) were also estimated. HOMO and LUMO’s energy gap of SH-29 was 0.114, and SK-119 was 0.109. Table 2. Based on distribution charges, HOMO and LUMO orbitals on compounds SH-29 and SK-119 were analyzed to identify the nucleophilic and electrophilic sites, whereas LUMO serves as an electron acceptor (electrophilic) and HOMO serves as an electron donor (nucleophilic). The orbitals (HOMO and LUMO) of compound SH-29 were distributed nearby the space on hydroxyphenyl’s benzene ring and imine bond, and in compound SK-119, HOMO and LUMO orbitals were distributed near the benzene ring and imine bond. Following this, MESP was also examined to calculate the electrostatic potential regions, physiochemical property, and hydrogen bonding interactions along with shape and size of ligands in terms of color, in which the blue color-coding region (signifying electron-deficient region) represents the most positive electrostatic potential and the red coding region (signifying electron-rich) represents the most electronegative potential of the molecules. The intermediate color-coding regions are orange, green, and yellow, and they prove the areas with electrostatic potentials close to zero. MESP map of the most electropositive region in compound SH-29 was found near the NH atom of imine portion and the hydrogen atoms of hydroxyl groups, and in the case of compound SK-119, it was observed with less potential positive effect. The most negative potentials were found around the oxygen atoms of the carboxyl group of SK-119 and for SH-29 around the oxygen atom of hydroxyl groups, although less profound. The 3D MESP analysis shows that the hydrogen bond donor and hydrogen bond acceptor site of SH-29 are found at the hydrogen atoms of hydroxyl groups and the oxygen atom of the phenyl moiety and imine nitrogen atom moiety, respectively. Whereas for SK-119, the vital activity comes from the carboxylic oxygen atoms of the oxopyrrolidine moiety.

### 3.4. In Vitro Nrf2 Activation Validation

Collectively, the above-mentioned in silico evaluation provides evidence that both SH-29 and SK-119 can bind to keap1 and potentially activate Nrf2. In order to validate these results, HaCaT keratinocytes were treated without or with 50 and 100 µM of the compounds. First, the cell viability was determined to exclude possible toxicity. As can be seen in Figure 5A, the compounds were well tolerated by the cells. In addition, no adverse effect was recorded by the vehicle (DMSO, 0.1%). Importantly, both compounds were able to activate Nrf2 (Figure 5B).

### 3.5. Pharmacological Activation of Nrf2 Reduce Air Pollution-Induced Damage

Diesel Particulate Matter (DPM) is used routinely to mimic several aspects of air pollution in vitro. The capacity of SH-29 and SK-119 to attenuate DPM-induced damage was monitored next. As expected, DPM caused a massive increase in ROS formation, which was reduced by 10 mM of N-acetyl cysteine (NAC) (commercial reference antioxidant), and the DMSO vehicle had no noticeable impact (Figure 6). Moreover, both compounds showed high efficacy and blocked ROS formation in a dose-dependent manner. SH-29 was more potent (IC_50_ of 2.3 in comparison to 11.3 of SK-119 µg/mL, calculated by four parameters regression in GraphPad prism), which also correlates to the Nrf2 activation capacity shown above. Concomitantly, the mitochondrial ROS generation was determined similarly. Once more, DPM enhanced its generation, whereas NAC attenuated it. SH-29 and SK-119 had comparable capacity to NAC but lower potency of those seen in cytosolic ROS.

Next, the impact of the compounds on cellular viability and DPM-induced inflammation was determined. DPM was able to enhance the secretion level of IL-8 by more than 4-fold. Importantly, both SH-29 and SK-119 completely revert the hyper-secretion of this inflammatory cytokine in all the tested concentrations. As both compounds have triphenolate moieties, it was possible that the reduction in ROS generation was due to direct scavenging capacity and not due to Nrf2 activation. Therefore, their direct action was investigated by the DPPH method. The results presented in the lower panel of Figure 6 suggest that, in the pharmacological range used above, no noticeable antioxidant capacity was seen. NAC was also used here as the positive control.

As both compounds are intended for topical usage, their safety in the human skin organ culture was investigated. The results in Figure 7 summarize the impact of the compounds on epidermal viability (A), skin irritation (B), and morphological alterations. As can be seen, 10% SDS (positive control) reduced epidermal viability in a significant manner, whereas both compounds at the same concentration were well tolerated by the tissue. In addition, IL-1α cytokine levels were not enhanced by the compounds in all the tested rang while SDS markedly enhanced its secretion by 5-fold. Lately, epidermal and dermal structures, shown by the hematoxylin eosin (H&E) staining, similarly suggest the safety of the topical application.

## 4. Discussion

Air pollution has been linked to several skin pathologies [8,13,30]. Despite significant improvements in air quality during the past decades, the majority of the world’s population continues to be exposed to levels of air pollution substantially above WHO Air Quality Guidelines [31,32]. During the past millennia, also high-income countries experienced severe air pollution problems and significant loss of disability-adjusted life-years [33,34]. Regulations on air pollution prevention and control were implemented addressing these threats, including the promulgation of the Clean Air Acts in the UK and USA [1,35]. As a result, measurable health benefits were achieved, as confirmed by long-term evaluation. These previous efforts provide useful insights and perspectives for air pollution prevention and control efforts in developing countries [36]. In general, it became a common understanding that monitoring and controlling the levels of air pollution contributed significantly to the improved air quality of high-income countries [37,38]. It was previously shown that control strategies limiting emissions of PM_2.5_ resulted in larger improvements in air quality compared to CO_2_ restrictions [39].

Aside from these control strategies, the recent shift towards renewable energy sources, in addition to the paradigm shift towards electrification within the transportation sector, will further improve air quality in high-income countries [40,41,42]. Albeit great progress of measures, controls, and improved air quality, the majority of the world’s population is still exposed to increased air pollution; thus, adverse effects of air pollutants on the skin remain a severe health risk that requires in-depth understanding [31]. The results of the current study demonstrate that pharmacological targeting the endogenous protection mechanism of the skin, e.g., Nrf2 master regular can reduce several aspects of air pollution include damage in vitro.

SK-119 and SH-29 share similar chemical properties. They both have triphenolate rings (at positions: 2, 4, 6) as electron-donating groups that form interactions with arginine- and serine-rich interface of Nrf2- Keap1. In addition, these groups could increase the solubility of the compounds. Moreover, both compounds share a benzyl core, which plays a hydrophobic key feature forming interaction with TYR 525 and PHE 572. SK-119 has one triphenolate ring, whereas SH-29 is a symmetric molecule with double triphenolate rings.

Additionally, both compounds (SH-29 and SK-119) resemble natural compounds reported to induce Nrf2 activation and showed beneficial actions in experimental in vivo and in vitro models, leading to the increased expression of antioxidant enzymes, including HO-1 and NQO1, and reducing oxidative stress markers [43]. Our previous work showed in silico calculation of several compounds with similar properties to SK-119 and SH-29, interacting with the arginine-rich area in the Kelch domain of Keap1 using the same mode as the Nrf2-ETGT motif [44]. In this work, our computational studies provided in-depth structural insight into the interacting residues of Keap1 with both SH-29 and SK-119. The molecular docking studies showed SH-29 has a higher docking score compared to SK-119, and these results were also validated in MM-GBSA analysis, showing lower total binding free energy for SH-29. Among all the interactions, the major contribution was of the lipophilic and the Van der Waals binding energies. Finally, HOMO, LUMO, and MESP analysis that were applied for both SH-29 and SK-119 to investigate the detailed aspects in terms of structure, electronics, and energy states showed that SH-29 has a smaller HOMO–LUMO energy gap which implies good stability terms of kinetic. These results of molecular modeling studies are in line with the experimental results showing both compounds were able to activate Nrf2, with SH-29 being a more potent Nrf2 activator.

Nrf2 is a pivotal player in the endogenous repair and antioxidant defense of the skin. Several studies showed that targeting this signaling pathway can reduce environmental stress-related dysfunctions. For instance, overexpression of Nrf2 protected against UV-Induced damage in a three-dimensional (3D) skin model [45], whereas Nrf2 null was severely affected by UVB [46]. Nrf2 activation was also shown to modulate skin immunity and local [47]. We and others have suggested that targeting its regulatory pathway can be used as a therapeutic option for several skin disorders [27,48]. Here, we provide evidence that pharmacological activation of Nrf2 can reduce air-pollution-induced damage in vitro.

DPM-induced damage is a widely used experimental system that mimics several key aspects of air pollution [49,50,51]. In dermal fibroblasts, DPM increases the levels of MMP-1, MMP-3, and TGFβ, resulting in extracellular alterations that are found in aged skin [52]. In keratinocytes, DPM was shown to increase ROS generation and mitochondrial damage [53,54] determined by DHR123 staining. Our data are in line with these findings and show similar results with MitoSOX fluorescent dye that specifically targeted mitochondria in living cells. SK-119 and SH-29 application reduced more effectively the cytosolic ROS formation than those in the mitochondria. Several reports demonstrated the regulatory role of ROS by Nrf2 in the mitochondria [55,56]. Interestingly, Nrf2-silence lung cells were shown to increase the susceptibility of the cells [57]. Although the protective effect of the compound was higher than N-acetylcysteine, the need for optimization in order to increase mitochondrial protection is still required.

Others showed reduced keratinocyte cellular viability resulting from increasing concentrations of DPM [58], while in our hand, this was not recorded. The discrepancy may be due to the use of serum-free conditions in the other study, resulting in higher bioavailability of the pollutants.

DPM was shown to increase skin inflammation. Zang et al. showed that HaCaT cells treated with 150 μg/mL PM_2.5_ increase the secretion level of both IL-1β and IL-6 in a microfluidic system [59]. Increased levels of IL-8, IL-1α, tumor Necrosis Factor-α (TNF-α), and Thymic Stromal Lymphopoietin (TSLP) were also reported by Li et al. [58]. Here, we found that the inflammatory cytokine IL-8 was upregulated by DPM and provide evidence that both direct antioxidant intervention (by NAC) or indirectly by Nrf2 activation will result in the inhibition of inflammation. This suggests that ROS overproduction by DPM also is one of the triggering pathways of inflammation [57].

The efficacy and safety of SH-29 and SK-119 were investigated in the well-established HaCaT keratinocyte cell line and in the ex vivo human skin organ culture systems [29,60,61,62,63]. Although the latter closely emulates the physical and biochemical properties of intact human tissue, it lacks the connection to the circulation, systemic inflammatory components, and the nervous system. Thus, further in vivo validation, as well as monitoring systemic adverse effects, are required. In addition, all our observations are on the short-term application of the compounds, and therefore, repeated application and long-term effects should also be evaluated.

## 5. Conclusions

Altogether, our results suggest that SK-119 and SH-29 can reduce the key aspect of air-pollution-induced skin disorders. In order to further evaluate the compound, a preliminary local safety assessment was performed in the ex vivo human skin organ cultures. This system was used as a valid experimental system for both efficacy and safety assessment with good correlation to clinical data [62,63,64,65]. Three biomarkers were tested and were unaltered by both compounds: skin morphology (evaluated by H&E staining), epidermal viability (MTT), and by monitoring the level of IL-1α, suggesting that the molecules are safer for topical usage. Further in vivo and clinical data are of need to validate these findings.

## 6. Patents

Gruzman A., Senderowitz H., Kahremany S., Cohen G. (2020) “New Substituted benzylidene-amino-phenyl-pyrrolidine-3-carboxylic acid derivatives and Uses Thereof” US Provisional Patent Application No. 62/813,052.

## Figures and Tables

**Figure 1 ijerph-18-12371-f001:**
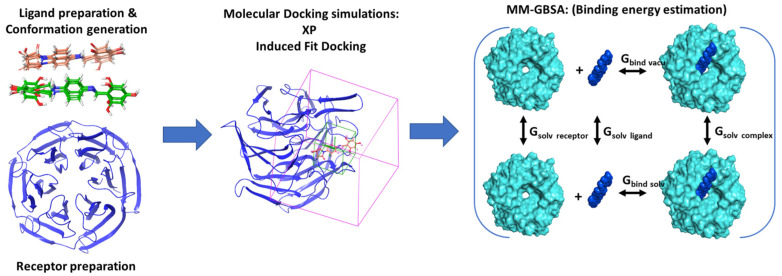
Designed workflow for in silico simulations of SH-29 and SK-119.

**Figure 2 ijerph-18-12371-f002:**
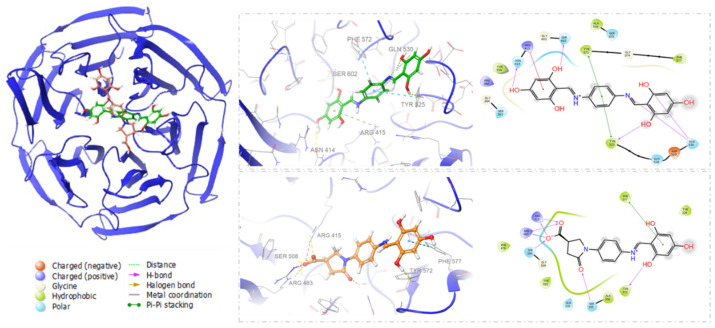
Docking pose of SH-29 (green) and SK-119 (orange) and their amino acid interactions with the binding site of Keap1. Ligands are represented in ball and stick.

**Figure 3 ijerph-18-12371-f003:**
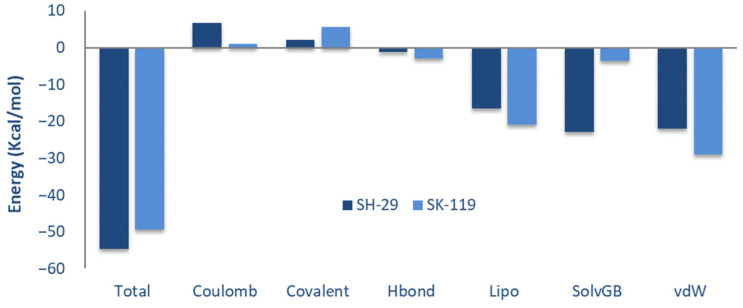
The total binding free energies of compound SH-29 (dark blue) and compound SK-119 (light blue) and the contribution of various interactions such as coulomb—coulomb energy, covalent—covalent binding energy, Hbond—H-bond energy, lipo—lipophilic energy, SolvGB—generalized born electrostatic solvation energy, vdw—Van der Waals energy.

**Figure 4 ijerph-18-12371-f004:**
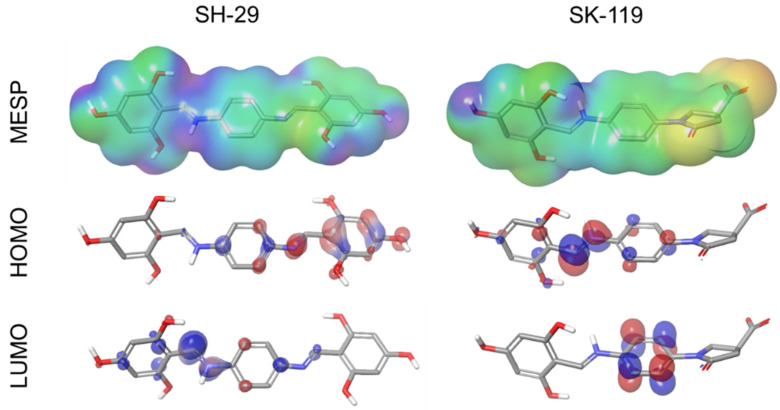
Representation of HOMO, LUMO, and MESP on compound SH-29 and compound SK-119.

**Figure 5 ijerph-18-12371-f005:**
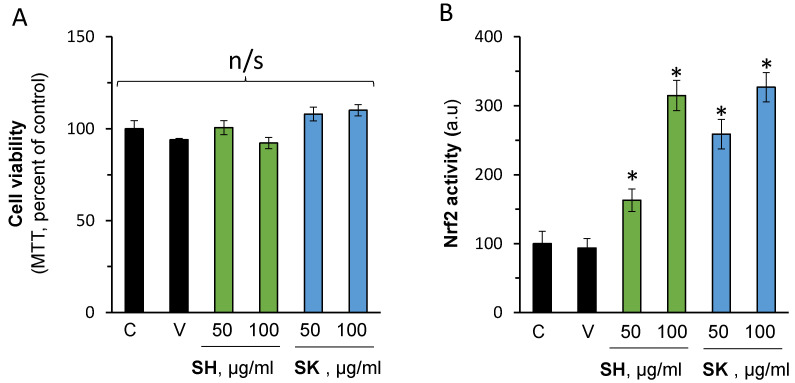
SH-29 and SK-119 activate Nrf2 in vitro. HaCaT keratinocyte cells were treated w/o or with 50 or 100 µg/mL of SH-29 or SK-119. After 2 h, cell viability was determined by the MTT assay (**A**). Concomitantly, the nuclear fraction was isolated, and Nrf2 activation was determined as written in the material and method section above (**B**). n = 3; * *p* < 0.05 for signification different from naïve control cells. C—control; V—vehicle, DMSO at 0.1%; SH for SH-29 and SK for SK-119.

**Figure 6 ijerph-18-12371-f006:**
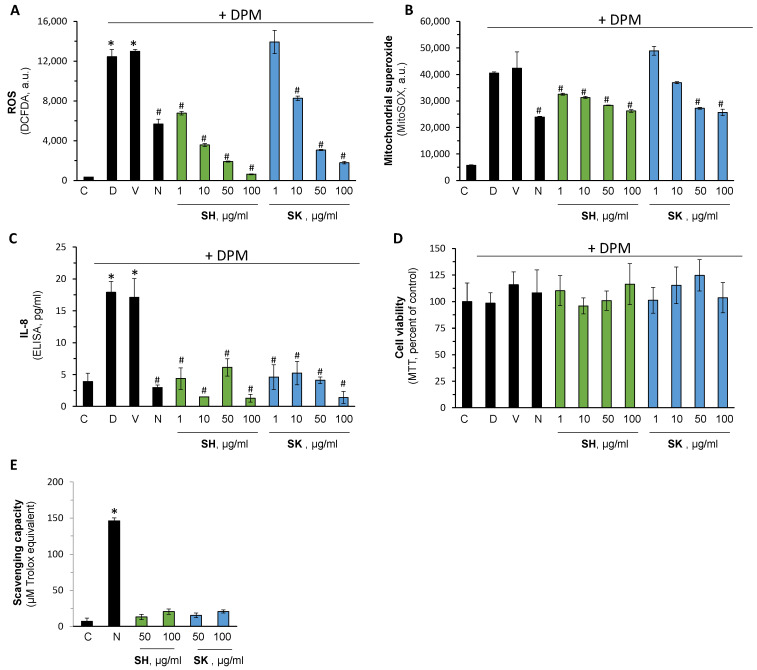
SH-29 and SK-119 attenuate DPM-induced damage. HaCaT cells were treated w/o or with the indicated SH-29 or SK-119 concentration and exposed to DPM. After 45 min., cytosolic (**A**) or mitochondrial (**B**) ROS levels were measured by DC-FDA or MitoSOX, respectively. After 24 h, IL-8 secretion (**C**) and cell viability (**D**) were determined by designated ELISA assay and MTT. In addition, the direct ROS scavenging capacity was evaluated by the DPPH method (**E**). n = 3; *^/#^
*p* < 0.05 for signification different from naïve control cells or DPM-stimulated group, respectively. C—control; D—DPM; V—vehicle, DMSO at 0.1%; N—N-acetylcysteine; SH for SH-29 and SK for SK-119.

**Figure 7 ijerph-18-12371-f007:**
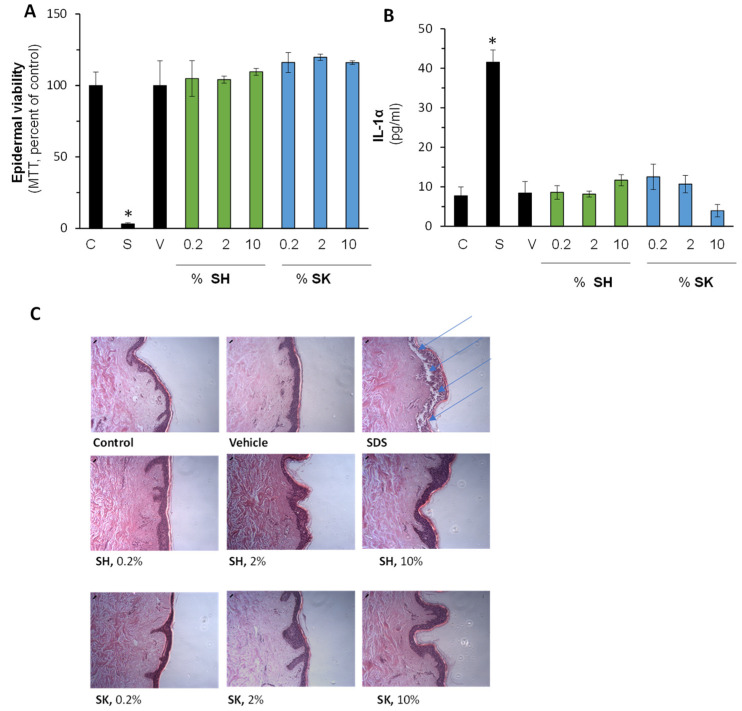
Safety evaluation of SH-29 and SK-119 in the ex vivo human skin organ culture. The compounds were applied topically on the epidermal layer of the skin samples in the indicated concentrations. After 24 h, epidermal viability was determined by the MTT assay (**A**). Concomitantly, IL-1α levels in the spent media were evaluated by ELISA (**B**). In addition, skin morphology after hematoxylin eosin (H&E) staining was assessed qualitatively (**C**). Blue arrows depict damaged (necrotic) tissue. n = 3; * *p* < 0.05 for signification different from naïve control tissues. SH-29 and SK-119 were tested at 0.2, 2 and 10% (*w*/*v*). C—control; V—vehicle, DMSO at 0.1%; S—SDS at 10%.

**Table 1 ijerph-18-12371-t001:** Binding free energy results on SH-29 and SK-119.

Compound	∆G_bind_ ^a^	∆G_coulomb_ ^b^	∆G_covalent_ ^c^	∆G_H-bond_ ^d^	∆G_sol lipo_ ^e^	∆G_sol GB_ ^f^	∆G_vdw_ ^g^
SH-29	−54.58	6.62	2.17	−1.27	−16.53	−22.94	−22.01
SK-119	−49.48	1.09	5.72	−2.86	−20.97	−3.49	−28.9

^a^ Free binding energy. ^b^ Coulomb energy contribution to the binding free energy. ^c^ Covalent energy contribution to the binding free energy. ^d^ H-bond energy contribution to the binding free energy. ^e^ The surface area due to lipophilic energy contribution to the binding free energy. ^f^ The generalized born electrostatic solvation energy contribution to the binding free energy. ^g^ Van der Waal’s energy contribution to the binding free energy.

**Table 2 ijerph-18-12371-t002:** Single point energy value of frontier orbital energies and electrostatic potential.

Compound	HOMO (ev)	LUMO (ev)	HOMO-LUMO GAP (ev)	MESP (kcal/mol)
SH-29	−0.222	−0.113	0.109	−6.94 to 150.47
SK-119	−0.127	−0.014	0.114	−167.78 to 39.83

## Data Availability

Publicly available datasets were analyzed in this study. This data can be found here: https://www.rcsb.org/; https://www.ncbi.nlm.nih.gov/, accessed on 10 October 2021.

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
