# Peer review of "SH-29 and SK-119 Attenuates Air-Pollution Induced Damage by Activating Nrf2 in HaCaT Cells"

_ijerph, 2021, doi:10.3390/ijerph182312371_

Round 1

Reviewer 1 Report

Manuscript ID: ijerph-1438464

Title: SH-29 and SK-119 attenuates air-pollution induced damage by activating Nrf2 in HaCaT cells

Authors: Kahremany et al.

The manuscript presents the influence of tSH-29 and SK-119, two activators of Nrf2 pathway, on diesel particulate matter-induced damage in human immortalized keratinocyte cell line (HACaT). The above manuscript seems to be interesting. However, the manuscript contains some errors and shortcomings that must be corrected before the possible acceptance of the manuscript for publication.

List of errors:

  1. Abstract, line 16 and next parts of the manuscript. All abbreviations should be presented in their full name at the point where they appear for the first time in the abstract and repeated in the body of the manuscript.
  2. Line 29, the second keyword. The authors probably wanted to write “Diesel Particulate Matter (DPM).
  3. Introduction, the first sentence, 36-37. The authors wrote that “In the last decade, a marked increase in indoor and outdoor airborne pollutants and 36 concomitant health-related problems has been observed”. This statement is supported by article published in 2009. In the last decade in the European Union and USA. New legal and technological solutions have been introduced that significantly reduced air pollution. Hybrid and electric cars have appeared, the consumption of coal for energy purposes is decreasing. Therefore, the authors should check the current changes in the level or air pollution in the last 10 years in the European Union, USA and Israel, and provide the current relevant references.
  4. Page 2 line 57. Replace “may lead to the collapse of the defense system and tissue damage” with “may lead to the collapse of the defense system and the development of tissue damage”.
  5. Page 2 and next pages. Material and Methods. For each reagent or equipment, the authors should provide their name (trade and generic), manufacturer/supplier name, city name and country name.
  1. Page 5, line 203. 2.3.1. Cell culture and ex vivo skin cultures. The authors stated that “The experiments were conducted with the approval of the IRB (Helsinki Committee) of Soroka Medical Center, Beer Sheva, Israel”. Apart a full name of “IRB”, the authors should provide the permit number, as well as the date of issue and expiry.
  2. Line 214, 221, 365 and other parts of the manuscript. The abbreviation of hour or hours is “h”, not “hr.”
  1. Line 218-@19. The statement “The epidermis sheets were then placed in 218 a 96-well plate and followed the method described above” is unclear and the method should be describe in detail.
  2. Discussion line 420-421, see comment 3.
  3. The cell line and skin sections are devoid the circulatory, immune, excretory, respiratory reproductive and nervous system. Therefore the study carried out on the cell line and skin sections do not allow to determine whether the tested substances disturb the function of these systems and lead to their damage. Besides, the observation made by authors are short-time. The effect of long-term exposure of tested compounds to cell line or skin section, and even more so to animals or humans, are unknown. These limitations of the research performed should be clearly stated at the end of the discussion and in the conclusions.

Author Response

The manuscript presents the influence of tSH-29 and SK-119, two activators of Nrf2 pathway, on diesel particulate matter-induced damage in human immortalized keratinocyte cell line (HACaT). The above manuscript seems to be interesting. However, the manuscript contains some errors and shortcomings that must be corrected before the possible acceptance of the manuscript for publication.

We thank the reviews for their time and suggestions that improved the manuscript

List of errors:

  1. Abstract, line 16 and next parts of the manuscript. All abbreviations should be presented in their full name at the point where they appear for the first time in the abstract and repeated in the body of the manuscript.

We thank the reviewer and added the full name that was missing in several places throughout the manuscript.

  1. Line 29, the second keyword. The authors probably wanted to write “Diesel Particulate Matter (DPM).

Indeed, thanks.

  1. Introduction, the first sentence, 36-37. The authors wrote that “In the last decade, a marked increase in indoor and outdoor airborne pollutants and 36 concomitant health-related problems has been observed”. This statement is supported by article published in 2009. In the last decade in the European Union and USA. New legal and technological solutions have been introduced that significantly reduced air pollution. Hybrid and electric cars have appeared, the consumption of coal for energy purposes is decreasing. Therefore, the authors should check the current changes in the level or air pollution in the last 10 years in the European Union, USA and Israel, and provide the current relevant references.

We thank the reviewer for his vital comment. We added the data from the last 10 years in the introduction.

“Despite significant improved air quality in high income countries due to agreements such as the Clean Air Act, air pollution is still one of the major environmental health risks. Chronic or repeated exposure to pollutants have been repeatedly associated to premature deaths and loss of millions disability-adjusted life-years around the world. Indeed, the European health agency still considers air pollution as the largest environmental health risk. Ultimately, causing cardiovascular and respiratory diseases that might lead to premature deaths.”

  1. Page 2 line 57. Replace “may lead to the collapse of the defense system and tissue damage” with “may lead to the collapse of the defense system and the development of tissue damage”.

We thank the reviewer and we amended as suggested.

  1. Page 2 and next pages. Material and Methods. For each reagent or equipment, the authors should provide their name (trade and generic), manufacturer/supplier name, city name and country name.

We thank the reviewer and added the missing information.

  1. Page 5, line 203. 2.3.1. Cell culture and ex vivo skin cultures. The authors stated that “The experiments were conducted with the approval of the IRB (Helsinki Committee) of Soroka Medical Center, Beer Sheva, Israel”. Apart a full name of “IRB”, the authors should provide the permit number, as well as the date of issue and expiry.

The missing data was integrated “ (#0258-19-SOR, approval protocol scrc20016; 29.06.2020).”  as this is a renewable permit (each year up to 5 years), we did not add the expiration date

  1. Line 214, 221, 365 and other parts of the manuscript. The abbreviation of hour or hours is “h”, not “hr.”

We thank the reviewer and we amended as suggested.

  1. Line 218-@19. The statement “The epidermis sheets were then placed in 218 a 96-well plate and followed the method described above” is unclear and the method should be describe in detail.

We agree with the review that the method lack details and amended it as follows

“Briefly, in order to detach the epidermis, the skin tissues were placed in PBS (56°c) for 1 min. Then, the epidermal layers were easily removed by gentle application of scalpel. The epidermis sheets were then placed in a 96-well plate containing the MTT solution. Following 1 h incubation at 37°C, the epidermal sheets were transfer to a new well containing isopropanol, and the eluted dye absorbance was measured by plate reader.”

  1. Discussion line 420-421, see comment 3.

We thank the reviewer and added the current improvement:

“Air pollution has been linked to several skin pathologies. Despite significant improvements of air quality during the past decades, the majority of the world’s population continues to be exposed to levels of air pollution substantially above WHO Air Quality Guidelines. During the past millennia, also high-income countries experienced severe air pollution problems and significant loss of disability-adjusted life-years. Regulations on air pollution prevention and control were implemented addressing these threats, including promulgation of the Clean Air Acts in the UK and USA. As a result, measurable health benefits were achieved, as confirmed by long-term evaluation. These previous efforts provide useful insights and perspectives for air pollution prevention and control efforts in developing countries. In general, it became common understanding that monitoring and controlling the levels of air pollution contributed significantly to the improved air quality of high income countries. It was previously shown that control strategies limiting emissions of PM2.5 resulted in larger improvements of air quality compared to CO2 restrictions.

Aside of these control strategies, the recent shift towards renewable energy sources in addition to the paradigm shift towards electrification within the transportation sector will further improve air quality in high income countries. Albeit great progress of measures, controls, and improved air quality, the majority of the world’s population is still exposed to increased air pollution, thus adverse effects of air pollutants on the skin remain a severe health risk that requires in-depth understanding.”

  1. The cell line and skin sections are devoid the circulatory, immune, excretory, respiratory reproductive and nervous system. Therefore, the study carried out on the cell line and skin sections do not allow to determine whether the tested substances disturb the function of these systems and lead to their damage. Besides, the observation made by authors are short-time. The effect of long-term exposure of tested compounds to cell line or skin section, and even more so to animals or humans, are unknown. These limitations of the research performed should be clearly stated at the end of the discussion and in the conclusions.

The study limitations are exactly as the reviewer pointed out, and now integrated in the discussion and conclusions section, as correctly suggested 

“The efficacy and safety of SH-29 and SK-119 were investigated in the well-established HaCaT keratinocyte cell line and in the ex vivo human skin organ culture systems. Although the latter closely emulates the physical and biochemical properties of intact human tissue, it lacks the connection to the circulation, systemic inflammatory components, and the nervous system. Thus, further in vivo validation, as well as monitoring systemic adverse effects are required. In addition, all our observations are on short-term application of the compounds, and therefore repeated application and long-term effect should also be evaluated.”

Reviewer 2 Report

The authors describe the synthesis and molecular characterization of two compounds SH-29 and SK-119 which were produced aiming to target Nrf2 signalling pathway. In silico simulations and molecular docking were performed as first assessment of the potential targets of these new molecules. In vitro and ex vivo data confirmed that these two compounds were able of protecting skin cells against air pollution. This is a very interesting work, with results that are relevant to the field and to a wide range of readers. However, the manuscript still has some points that need clarification and correction.

In figure 5 authors show that SK-119 at 50 mM is more effective in stimulating Nrf2. Did authors test the concentrations of 1 and 10 mM? From Figure 5, SK-119 seems to have a better potential in protecting the cells against ROS through Nrf2 activation, however in Figure 6, when measuring the ROS content of cells and mitochondria, SH-29 shows a better performance. Do authors have an explanation for this?

Figure 7. Why using a lower range of SH-29 and SK-119 concentrations on ex vivo assays compared to the in vitro? Here the highest concentration was 10 mM instead of 100 mM as in HaCaT cells? Is there a reason?

Line 39, in the ozone molecular formula write the 3 as subscript. Also write correctly the formulas in lines 87,

Line 54, please define ROS (although it is a common abbreviation, all abbreviations should be defined at first appearance). Check the whole document and define all abbreviations not defined.

Line 104-105, the sentence seems incomplete. Please revise.

SI units are recommended, use h to abbreviate hour and not hr or hr., the same abbreviation should be use along the manuscript, the SI unit is recommended. The three forms are used in the same paragraph, lines 214, 221 and 223.

Title of section 2.3.4, should be changed to a more concise one. As for example, “in vitro scavenging activity”, in vitro antioxidant activity”,

Line 243-244, please indicate the conditions of incubation

Figure 3. Authors should describe in the legend the meaning of “Lipo”, “SolvGB” and “vdW”, as they may not be understandable to all readers.

Line 324-325, for negative numbers do not leave a space between the minus symbol and the value.

Line 347, correct the word “oxygans”, should be “oxygen”

Figure 6, the authors need to include in the legend the meaning of “D”.

Line 374, add the abbreviation NAC to N-acetyl cysteine. As in line 379 the NAC abbreviation appears without pre-definition.

In section 3.5, author show the values of EC50 (usually defined as Half maximal effective concentration), did they want to report half maximal inhibitory concentration (IC50)? In my opinion IC50 better applies to this data, however I would like to hear the authors. Also, authors should include in methods an explanation of how they calculated the EC50 shown in line 377. If values were normalized or not, in case of normalization what was the reference, and so on.

Figure 7. include in its legend the meaning of “S”, “C” and “V”. Figure legends should be read alone, without the need to search in the text for the meaning of abbreviations.

Line 408, please add the definition of H&E.

Supplementary material

Page S2, revise the structure of itaconic acid.

Author Response

The authors describe the synthesis and molecular characterization of two compounds SH-29 and SK-119 which were produced aiming to target Nrf2 signalling pathway. In silico simulations and molecular docking were performed as first assessment of the potential targets of these new molecules. In vitro and ex vivo data confirmed that these two compounds were able of protecting skin cells against air pollution. This is a very interesting work, with results that are relevant to the field and to a wide range of readers. However, the manuscript still has some points that need clarification and correction.

 We thank the reviews for their time and suggestions that improved the manuscript

In figure 5 authors show that SK-119 at 50 mM is more effective in stimulating Nrf2. Did authors test the concentrations of 1 and 10 mM? From Figure 5, SK-119 seems to have a better potential in protecting the cells against ROS through Nrf2 activation, however in Figure 6, when measuring the ROS content of cells and mitochondria, SH-29 shows a better performance. Do authors have an explanation for this?

The expert reviewer is of course correct, in both the in-silico and ROS efficacy measurements SH-29 showed superior action, but in the Nrf2 activation assay, the same efficacy was seen but with lower (not significant) potency, determined by the “50” point. Our initial thought was to use the nrf2 activation assay only to validate the in silico results. In retrospect, the measurement of additional concentrations would have enabled us to clarify and compare the exact ability of the compounds, and to determine the correlation between nrf2 activation and its ameliorating effect. As a results, we have only stated in the text “Importantly, both compounds were able to activate Nrf2” as the results cannot claim comparison between the compounds in this experimental setting.

Figure 7. Why using a lower range of SH-29 and SK-119 concentrations on ex vivo assays compared to the in vitro? Here the highest concentration was 10 mM instead of 100 mM as in HaCaT cells? Is there a reason?

The concentrations tested in the ex vivo skin are presented as percent active compound, as most topically applied compounds (and are actuality much higher than tested in the cells). to clarify it,  the, we have emphasized it in the figure legend.

Line 39, in the ozone molecular formula write the 3 as subscript. Also write correctly the formulas in lines 87,

We thank the reviewer, and corrected the formulas.

Line 54, please define ROS (although it is a common abbreviation, all abbreviations should be defined at first appearance). Check the whole document and define all abbreviations not defined.

We thank the reviewer, and added all the abbreviations.

Line 104-105, the sentence seems incomplete. Please revise.

We thank the reviewer and corrected the sentence.

SI units are recommended, use h to abbreviate hour and not hr or hr., the same abbreviation should be use along the manuscript, the SI unit is recommended. The three forms are used in the same paragraph, lines 214, 221 and 223.

We thank the reviewer and amended as suggested.

Title of section 2.3.4, should be changed to a more concise one. As for example, “in vitro scavenging activity”, in vitro antioxidant activity”,

We thank the reviewer and changed the title to his second option.  

Line 243-244, please indicate the conditions of incubation

Added:

“Then, the cells were incubated at 37°C with DPM (in PBS) at a final concentration of 100 µg/ml in the absence or presence of the compounds (tested at 1-100 µg/ml).”

Figure 3. Authors should describe in the legend the meaning of “Lipo”, “SolvGB” and “vdW”, as they may not be understandable to all readers.”

We thank the reviewer and added the meaning for all components.

Line 324-325, for negative numbers do not leave a space between the minus symbol and the value.

We thank the reviewer, and we corrected the numbers.

Line 347, correct the word “oxygans”, should be “oxygen”

We thank the reviewer and amend the term.

Figure 6, the authors need to include in the legend the meaning of “D”.

Thanks, “D- DPM” was added

Line 374, add the abbreviation NAC to N-acetyl cysteine. As in line 379 the NAC abbreviation appears without pre-definition.

We thank the reviewer and added the term.

In section 3.5, author show the values of EC50 (usually defined as Half maximal effective concentration), did they want to report half maximal inhibitory concentration (IC50)? In my opinion IC50 better applies to this data, however I would like to hear the authors. Also, authors should include in methods an explanation of how they calculated the EC50 shown in line 377. If values were normalized or not, in case of normalization what was the reference, and so on.

EC50 values were calculated by four parameters (variable slop, non-linear) regression in GrafPad prism (added to the manuscript). The data was not normalized or transformed to logarithmic scale prior to analysis. Our initial thought was to use EC50 nomenclature and not IC50 as the readers can be misled to think that as the compounds are inhibitors and not activators. However, as the analysis was indeed performed on the values (descending), and not on the precent of ROS inhibition (positive and increasing value), we changed according to the reviewer’s suggestion.

Figure 7. include in its legend the meaning of “S”, “C” and “V”. Figure legends should be read alone, without the need to search in the text for the meaning of abbreviations.

The reviewer is correct. Changes as suggested

Line 408, please add the definition of H&E.

Added, as well as more details about the hematoxylin eosin staining In the method section that were missing

Supplementary material

Page S2, revise the structure of itaconic acid.

We thank the reviewer, both structures of itatonic acid are acceptable and we chose the second option shown below. We also rearranged the legend to “itaconic acid (1), phenylenediamine (2)”.

Round 2

Reviewer 1 Report

Manuscript ID: ijerph-1438464

Title: SH-29 and SK-119 attenuates air-pollution induced damage by activating Nrf2 in HaCaT cells

Authors: Kahremany et al.

In the opinion of the reviewer, the manuscript is ready for publication.

Reviewer 2 Report

The authors have answered all the questions and corrected/clarified the manuscript.  In my opinion the manuscript is ready to be published.